The effects of the E3 ubiquitin–protein ligase UBR7 of Frankliniella occidentalis on the ability of insects to acquire and transmit TSWV

Shi Junxia 1 2
Zhou Junxian 3
Jiang Fan 2
Li Zhihong 1 lizh@cau.edu.cn
Zhu Shuifang 1 2 zhusf@caiq.org.cn
1 MARA Key Laboratory of Surveillance and Management for Plant Quarantine Pests, College of Plant Protection, China Agricultural University , Beijing , China
2 Institute of Plant Quarantine, Chinese Academy of Inspection and Quarantine , Beijing , China
3 Agricultural Technology Service Center of Yunyang County , Chongqing , China
Sotelo-Mundo Rogerio
Electronic publication date: 2023 May 9
Publication date: 2023
Volume: 11
Electronic Location ID: e15385
Received 2022 Dec 21; Accepted 2023 Apr 18
Copyright: © 2023 Shi et al.
Copyright year: 2023
Copyright holder: Shi et al.
License: This is an open access article distributed under the terms of the Creative Commons Attribution License, which permits unrestricted use, distribution, reproduction and adaptation in any medium and for any purpose provided that it is properly attributed. For attribution, the original author(s), title, publication source (PeerJ) and either DOI or URL of the article must be cited.
License URL: https://creativecommons.org/licenses/by/4.0/

Keywords: Frankliniella occidentalis, Tomato spotted wilt orthotospovirus, E3 ubiquitin–protein ligase, UBR7, Transmission efficiency

Funding: National Key Research and Development Program of China 2021YFF0601901; 2019YFC1604704; 2016YFC1200605 This work was supported by the National Key Research and Development Program of China (NOs. 2021YFF0601901; 2019YFC1604704; 2016YFC1200605). The funders had no role in study design, data collection and analysis, decision to publish, or preparation of the manuscript.

==============================
The interactions between plant viruses and insect vectors are very complex. In recent years, RNA sequencing data have been used to elucidate critical genes of Tomato spotted wilt ortho-tospovirus (TSWV) and Frankliniella occidentalis (F. occidentalis). However, very little is known about the essential genes involved in thrips acquisition and transmission of TSWV. Based on transcriptome data of F. occidentalis infected with TSWV, we verified the complete sequence of the E3 ubiquitin-protein ligase UBR7 gene (UBR7), which is closely related to virus transmission. Additionally, we found that UBR7 belongs to the E3 ubiquitin–protein ligase family that is highly expressed in adulthood in F. occidentalis. UBR7 could interfere with virus replication and thus affect the transmission efficiency of F. occidentalis. With low URB7 expression, TSWV transmission efficiency decreased, while TSWV acquisition efficiency was unaffected. Moreover, the direct interaction between UBR7 and the nucleocapsid (N) protein of TSWV was investigated through surface plasmon resonance and GST pull-down. In conclusion, we found that UBR7 is a crucial protein for TSWV transmission by F. occidentalis, as it directly interacts with TSWV N. This study provides a new direction for developing green pesticides targeting E3 ubiquitin to control TSWV and F. occidentalis.

Introduction

Tomato spotted wilt orthotospovirus (TSWV), a member of the order Bunyavirales, family Tospoviridae and genus Orthotospovirus, was first discovered by Brittlebank in Australia in 1915 (Brittlebank, 1919). In the last decade, due to climate change, human activities, agricultural production, and arthropod spread, the incidence of TSWV has been increasing (Liang, Yuan & Li, 2020; Mandal et al., 2007; Sivaprasad et al., 2018). TSWV can infect over 1,060 plants in 85 families (Parrella et al., 2003). Because of the significant damage, TSWV is listed as one of the world’s ten most harmful plant viruses (Scholthof et al., 2011). Consequently, the European and Mediterranean Plant Protection Organization (EPPO) classifies it as an A2 quarantine pathogen.

TSWV is only spread by thrips (Ullman et al., 2002), including western flower thrips (Frankliniella occidentalis), flower thrips (F. intonsa), palm thrips (Thrips palmi), and tobacco thrips (T. alliorum). The western flower thrips (F. occidentalis (Pergande)) is the most efficient species for the transmission of TSWV and as such, has attracted much attention. Because of its small size, good concealment, agile action, rapid reproduction, broad host range, and high insecticide resistance, F. occidentalis has brought severe economic losses to many countries and regions (Gupta, Kwon & Kim, 2018). China has included F. occidentalis in the IAS1000 Project (a genome project comprising 1,000 invasive alien species).

Like most Tospoviridae, TSWV replicates in its insect vectors and is transmitted in the vector’s saliva during persistent feeding (Picó, Díez & Nuez, 1996; Wijkamp et al., 1995). F. occidentalis is primarily infected by feeding on plants during the first instar larval stage. During the transmission of TSWV by F. occidentalis, TSWV needs to replicate and break through the vector’s multiple infection barriers to reach the salivary glands of the thrips (Chen et al., 2020; Hogenhout et al., 2008). TSWV first infects the midgut epithelial cells of F. occidentalis (Gupta, Kwon & Kim, 2018; Montero-Astúa, Ullman & Whitfield, 2016). After TSWV infects thrips, it replicates in multiple tissues, similar to the mode of other Bunyavirales virus infections (Whitfield, Ullman & German, 2005). Furthermore, TSWV proliferates in the salivary glands and other tissues of F. occidentalis (Ullman et al., 1993; Wetering, Goldbach & Peters, 1996). The route of virus infection is transferred from the midgut and tubular salivary glands in the larval stage to primary salivary glands in the adult stage (Montero-Astúa, Ullman & Whitfield, 2016). Infected adult insects carry the virus for life and spread it by feeding on healthy plants, thereby infecting them (Wetering, Goldbach & Peters, 1996). Therefore, identifying the key proteins that affect the transmission of TSWV by F. occidentalis is crucial to understanding the relationship between pathogens and their hosts. This interaction is also the basis for the control measures of TSWV and its transmission vector, F. occidentalis.

The transcription, replication and viral particle assembly processes are similar for viruses entering the host cell, both in plant and insect cells (Wang et al., 2019). The study of Bunyavirales viruses in infected animals by reverse genetic system revealed that the viral RNA-dependent RNA polymerase (RdRp) is responsible for the transcription of mRNA in the replication cycle of viruses; it steals the host mRNA5 ‘cap and replicates the RNA genome through the “CAP-snatching” mechanism (Elliott, 2014). The viral nucleocapsid protein N is closely related to the formation of ribonucleoproteins transcription and replication of viral genome (Yu et al., 2015). Thus, viruses are believed to transcribe, replicate and assemble TSWV viral particles in mesophilic cells through the RdRp and N proteins. Numerous studies on virus-host interactions demonstrate that the viral N protein plays a critical role (Liu et al., 2015; Ganaie & Mir, 2014; Zu et al., 2019). However, whether the TSWV N protein interacts with F. occidentalis and is associated with the virus transmitted by thrips needs to be clarified.

Meanwhile, little is known about thrips’ response to TSWV during infection from larval acquisition to adult inoculation of plants. RNA sequencing (RNA-seq) is a powerful tool for identifying differentially expressed transcripts during host–pathogen interactions. Therefore, several RNA-seq-based transcriptome analyses have been performed to determine the differentially expressed genes in TSWV-infected F. occidentalis (Schneweis, Whitfield & Rotenberg, 2017; Shrestha et al., 2017; Zhang et al., 2013). De novo transcriptome sequencing identified 278 unigenes genes involved in plant virus transmission and insecticide resistance (Zhang et al., 2013). Schneweis, Whitfield & Rotenberg (2017) used RNA-seq to determine the overall transcriptome response of first instar larvae, pupae, and adults of F. occidentalis to viral infection. The differentially expressed putative innate immunity-related transcript genes included zinc finger protein, hexamerin 2 B, and thyroid peroxidase precursor (Schneweis, Whitfield & Rotenberg, 2017). The transcripts that responded to TSWV differed during various developmental stages, reflecting the relationship between thrips development and virus transmission via insect vectors and the coordination between development and the virus dissemination route (Schneweis, Whitfield & Rotenberg, 2017; Shrestha et al., 2017). Transcriptomic and network integration analysis of the larval gut of F. occidentalis found that zinc finger proteins were associated with TSWV infection (Han & Rotenberg, 2021).

Additionally, we referred to the primary sequence of a draft genome (https://doi.org/10.15482/USDA.ADC/1503960) of F. occidentalis and three developmental-stage transcriptomes to identify TSWV responsive genes (Schneweis, Whitfield & Rotenberg, 2017). As inferred from the Gene Ontology (GO), TSWV infection interferes with host defenses, insect cuticle structure, development, metabolism, and transport processes and functions. Moreover, most of these studies are at the level of sequencing and prediction, and there are few basic biological studies on the interaction between TSWV and F. occidentalis. Badillo-Vargas et al. (2019) identified six interacting proteins of the glycoprotein GN by blot overlay assays, among which cyclophilin and endoCP-GN could directly interact with GN. Wan et al. (2019) successfully constructed a yeast two-hybrid library and TSWV membrane protein GN and GC bait vector. Subsequently, they screened proteins interacting with TSWV GN, including ubiquitin-related proteins (Zheng et al., 2020). These studies suggest a possible interaction between TSWV and F. occidentalis at the protein level.

Ubiquitin, one of the most common posttranslational protein modifiers, is widely and highly conserved in all eukaryotes. The ubiquitination process, consisting of a tertiary enzyme-linked reaction consisting of an E1 ubiquitin-activating enzyme, an E2 ubiquitin-conjugating enzyme, and an E3 ubiquitin–protein ligase, is an essential posttranslational modification (PTM) protein degradation process and a crucial aspect of protein interaction between host and pathogen (Alejandro et al., 2019; Schinz & Littlefield, 1985). E3 ubiquitin–protein ligase is responsible for protein-specific ubiquitination and promotes ubiquitin transfer by producing E2 ubiquitin–protein conjugating enzyme. E3 is the critical factor in the last step of the ubiquitination cascade (Alejandro et al., 2019; Schinz & Littlefield, 1985). Through the interaction with specific substrates, this enzyme determines spatiotemporal properties and the pathway specificity (Spratt et al., 2012). Many E3 ubiquitin–protein ligase members have been reported to degrade human/animal virus proteins, thus negatively affecting virus accumulation in host cells (Wang et al., 2021; Yang et al., 2019). Viruses use host cells to synthesize proteins that influence and control the host, and thus protein expression in the host is conducive to viral proliferation (Snippe, Goldbach & Kormelink, 2005). The host’s E3 ubiquitin–protein ligase system can selectively degrade viral proteins, restrict viral growth, and bring the virus and host to homeostasis. (Tang et al., 2018). To counter the antiviral mechanism of the host ubiquitin–proteasome system (UPS), the viruses have evolved strategies to use or destroy the UPS to inactivate or degrade cellular proteins and thereby promote viral reproduction (Tang et al., 2018).

E3 ubiquitin–protein ligases are divided into three subfamilies according to the characteristic domain and the mechanism of ubiquitin transfer: the RING (including the U-box protein family), HECT (homologous to the E6AP C-terminus), and RING-in-related-RING (RBR) family (Hatakeyama, 2017). Although E3 ubiquitin–protein ligases from different families can catalyze the covalent connection between ubiquitin and lysine residues in target proteins, the mechanisms are different (Berndsen & Wolberger, 2014; Mattiroli & Sixma, 2014; Zheng & Shabek, 2017). RING-type E3s are the most common E3 ubiquitin–protein ligases (Deshaies & Joazeiro, 2009), related to DNA repair and immune signaling pathways (Hatakeyama, 2017).

The UBR7 protein of the E3 ubiquitin–protein ligase family, belonging to the RBR subfamily, encodes a UBR box-containing protein (zinc finger in N-recognin) and contains a plant homeodomain (PHD) in the C-terminus (Dasgupta et al., 2022). UBR7 plays a role in the N-terminal rule proteolytic pathway that is highly conserved in yeast, animals, and plants (Lee et al., 2008; Zimmerman et al., 2014). A study based on TurboID used proximity labeling demonstrated that the UBR7 protein in the E3 ubiquitin–protein ligase family directly interacts with the N Toll-interleukin-1-receptor domain (TIR) of the nucleotide-binding leucine-rich repeat (NLR) immune receptors (Zhang et al., 2019). NLR can recognize viral proteins upon infection and trigger RNA silencing-based antiviral immunity, one type of plant immunity. Hence, UBR7 downregulation led to increased N protein and enhanced TMV resistance (Zhang et al., 2019). So we wondered if insects also have a protein similar to plant UBR7 involved in autoimmunity. There have been many studies on the function of E3 ubiquitin ligase in plant autoimmune regulation, Still, the process function of UBR7 in insects is not straightforward, and no relevant studies have been conducted in thrips.

Here, we verified the UBR7 gene, a protein closely related to TSWV, from F. occidentalis using an existing data (Schneweis, Whitfield & Rotenberg, 2017). We detected the expression of the UBR7 gene in different F. occidentalis instars and explored the effect of UBR7 on the acquisition and transmission of TSWV. Surface plasmon resonance (SPR), GST pull-down, and Co-IP assays were used to demonstrate the direct interaction between the UBR7 protein of F. occidentalis and the TSWV N protein. We aimed to provide insights for the research and development of novel molecules for the simultaneous targeted control of TSWV and thrips.

Materials and Methods

Transcriptome sequence analysis of F. occidentalis in response to TSWV infection

The transcriptome data of F. occidentalis response to TSWV were obtained from the National Center for Biotechnology Information (NCBI) Sequence Read Archive (SRA) database (see Table S1 for GenBank accession number) (Schneweis, Whitfield & Rotenberg, 2017).

According to the analysis method of previous studies (Schneweis, Whitfield & Rotenberg, 2017), the transcriptome of TSWV-infected and TSWV-free F. occidentalis at the same developmental stage was analyzed to obtain the differentially expressed genes among the six groups and the differentially expressed genes were ranked according to the fold_change value (Table S2). However, in this transcriptome study of the interaction F. occidentalis with TSWV, the authors have verified seven up-regulated or down-regulated genes by real-time quantitative PCR (RT-qPCR) (Schneweis, Whitfield & Rotenberg, 2017). Thus, we did not repeat another RT-qPCR validation again for these transcriptome data.

Plant materials and TSWV inoculum

Datura stramonium plants were kept in a growth chamber at 25 °C with a 16-h light/8-h dark photoperiod.

TSWV was a gift from Nanjing Agricultural University. This virus was maintained on Datura stramonium plants (Wan et al., 2018). TWSV-infected plant tissue for the acquisition access period (AAP) was obtained by mechanical inoculation of 3-week-old D. stramonium plants or through F. occidentalis insect vectors (Zhao et al., 2016). Infected tissue was ground in a chilled mortar and pestle in 10 mL of general extract buffer (Agdia, Elkhart, IN, USA). Plants to be inoculated were dusted with carborundum and gently rubbed with a cotton swab wetted with inoculum. A total of 12 days after mechanical inoculation, the leaves appeared deformed, curled, chlorotic, stained and displayed other symptoms.

Insect culture

Frankliniella occidentalis, a susceptible laboratory strain, was a gift from the Institute of Vegetables and Flowers, Chinese Academy of Agricultural Sciences. The F. occidentalis colony was maintained on green beans (Phaseolus vulgaris) at 25 °C, 50% relative humidity, and a 16-h light/8-h dark photoperiod (Zhao et al., 2016).

Fresh beans were placed in the insect cages for the adult thrips to lay eggs. After 3 days, the thrips were removed with a brush, and the beans were placed in a new cage. The larvae were allowed to incubate and were fed the leaves of D. stramonium with or without TSWV for 72 h and then fed with green beans until they emerged as adults. First instar larvae (L1s), second instar larvae (L2s), pupae, female adults, and male adults of thrips were collected according to the methods of previous studies (Sheida et al., 2016; Zhi, Li & Gai, 2010). The head, thorax and abdomen of thrips were dissected under a dissecting microscope (Nikon, Tokyo, Japan). Subsequently, these samples were collected for total RNA and extraction and total protein extraction. The experiments were repeated six times for each developmental stage (n ≥ 6).

Cloning of the UBR7 gene

RNA isolation and first-strand cDNA synthesis

The total RNA of adult F. occidentalis with TSWV was extracted using TransZol Up Reagent (TransGen, Beijing, China). Then, RNA integrity was further affirmed using agarose gel electrophoresis. Finally, genomic DNA elimination and reverse transcription were conducted using TransScript® II One-Step gDNA Removal and a cDNA Synthesis SuperMix kit (TransGen, Beijing, China). The synthesized cDNA was stored at −20 °C for further use.

Cloning of the full-length UBR7 cDNAs

Based on transcriptome data analysis and previous studies (Schneweis, Whitfield & Rotenberg, 2017), we focused on UBR7 (XM_026422690.1). First, the primer sequences were designed using Primer Premier 5.0 software to verify the full-length UBR7 gene (Table 1). Then, according to the instructions of a 2 × TransTaq® High Fidelity PCR SuperMix I (-dye) kit (TransGen, Beijing, China), PCR amplification was conducted in an Applied Biosystems Veriti™ Dx 96-Well Fast Thermal Cycler (Thermo Fisher, Waltham, MA, USA). The purified PCR products were then ligated into the pClone007 Vector (Tsingke, Beijing, China) and were transformed into Trans5α Chemically Competent Cells (TransGen, Beijing, China). Positive clones were selected by ampicillin resistance and then sequenced by Tsingke Biotechnology, Beijing, China.

Table 1 The primer pairs for RT-PCR used in this study.

Target gene	Primer	Species	Primer sequence	Correlation coefficient	PCR efficiency	Application	Amplicon size	Annealing temperature	
(5′ to 3′)	(R2)	(%)	(bp)	(°C)	
UBR7	UBR7-F	Frankliniella occidentalis	GAAAACAACCAGTCAACGAA	/	/	PCR	1889	55	
UBR7	UBR7-R	TAGCCACCACCATCAAAAC	
UBR7	ds-UBR7-F	Frankliniella occidentalis	TAATACGACTCACTATAGGG
TGTGAACAGTGCATGAGCAA	/	/	dsRNA synthesis	310	58	
UBR7	ds-UBR7-R	TAATACGACTCACTATAGGG
GCCAAAATGTTGCTCCCTTA	
EGFP	ds-EGFP-F	Aequorea victoria	TAATACGACTCACTATAGGG
CAGTGCTTCAGCCGCTAC	/	/	dsRNA synthesis	287	58	
EGFP	ds-EGFP-R	TAATACGACTCACTATAGGGG
TTCACCTTGATGCCGTTC	
UBR7	q-UBR7-F	Frankliniella occidentalis	CAGATGATGACGAAAGTAACGC	0.998	105.3	q-PCR	209	55	
UBR7	q-UBR7-R	AGCAAGGCATACACCACCTC	
Actin	q-Actin-F	Frankliniella occidentalis	GGTATCGTCCTGGACTCTGGTG	1.000	102.8	q-PCR	149	55	
Actin	q-Actin-R	GGGAAGGGCGTAACCTTCA	
TSWV N	q-TSWV-F	Tomato spotted wilt orthotospovirus	CTTGCCATAATGCTGGGAGGTAG	0.990	100.8	q-PCR	112	58	
TSWV N	q-TSWV-R	TCCCGAGGTCTTTGTATTTTGC	

Gene characterization and phylogenetic analysis

To fully understand UBR7, we performed a bioinformatics analysis. First, UBR7 gene sequence analysis was performed using the NCBI Basic Local Alignment Search Tool (BLAST) program. ScanProsite was used to identify possible functional sites (Castro et al., 2006). The deduced protein sequences’ molecular weights and isoelectric points were computed using the ExPASy Proteomics Server. The physicochemical properties of UBR7 were evaluated through Protparam. To analyze the sequence homology and phylogenetic relationships of UBR7, E3 ubiquitin-protein ligase gene information from different species (including Thrips palmi, Cryptotermes secundus, Pediculus humanus corporis, Nilaparvata lugens, and Laodelphax striatellus) was downloaded from GenBank. Then, a phylogenetic tree was constructed using the neighbor-joining method with 1,000 bootstrap replicates through MEGA 5.0 (Nei et al., 2013).

Studies have shown that the UBR7 gene is involved in the antiviral immune mechanism of plants (Tang et al., 2018; Zhang et al., 2019). Thus, we compared the amino acid sequences (AAs) of the UBR7 protein from TSWV-susceptible plants with the UBR7 protein from F. occidentalis to understand the conservation of the UBR7 protein. The AAs were downloaded from NCBI GenBank (including Solanum, Nicotiana, and Datura) for the multiple sequence alignment of conserved regions of UBR7. The AAs were analyzed with Clustal and T-coffee to visualize the conserved motifs. In addition, WebLogo was employed to illustrate the amino acid frequencies.

UBR7 gene interference

dsRNA synthesis

RNA interference (RNAi) primers were designed using Harvard’s SnapDragon program based on the UBR7 gene sequence. The T7 promoter sequence (-TAATACGACTCACTATAGGG-) was added to the 5′ ends of the forward and reverse primers. After pretest screening, we selected the dsRNA primer with the highest interference efficiency (Table 1). The PCR was then carried out using ds-UBR7 primers and the negative control EGFP (ds-EGFP) primers by referring to previous studies (Table 1) (Xiang et al., 2011). The DNA template of the experimental group was from F. occidentalis, and the negative control group was EGFP plasmid (Beyotime, Shanghai, China). The T7 RiboMAX™ Express RNAi System (Promega, Madison, WI, USA) synthesized dsRNA. After measuring the concentration, the samples were stored at −80 °C.

dsRNA feed preparation

Sucrose was weighed and dissolved in DEPC water to prepare a solution with a 30% mass fraction as a solute of artificial feed. Then, the sucrose solution was used to dilute the dsRNA to the concentration of 500 ng/μL. Three diets groups were subsequently prepared (blank control, CK: 30% sucrose + ddH2O; negative control group, ds-EGFP: 30% sucrose + dsRNA-EGFP; and experimental group, ds-UBR7: 30% sucrose + dsRNA-UBR7).

Interference efficiency and survival rate detection

The entire RNAi method and apparatus were referenced from previous reports (Wu et al., 2018). Adult thrips 3 days after eclosion without TSWV were randomly selected regardless of sex. The thrips were divided into three groups, CK, ds-EGFP, and ds-UBR7, and placed in separate plastic cups, and the experiment was repeated three times. The thrips were starved for 4 h, then fed an artificial diet. Twenty-four hours later (Wu et al., 2018; Yuan, 2019), the interference efficiency of the UBR7 gene in F. occidentalis was examined by RT-qPCR, and the survival rate of F. occidentalis was determined.

Gene expression and viral abundance detection

A total of 3 days after eclosion, adult thrips carrying TSWV (V+) were randomly selected irrespective of sex and divided into three groups (CK, ds-EGFP, and ds-UBR7). The thrips were fed artificial feed after 4 h of starvation treatment. The samples were collected after 0, 6, 12, 24, 48, 72, and 96 h, with three replicates of 50 thrips in each samples. The expression of the UBR7 gene and the TSWV abundance in thrips were then detected via RT-qPCR. No additional artificial feed was supplemented throughout the experimental process. The entire process is illustrated in Fig. 1A.

Figure 1 Effects of UBR7 RNA interference on UBR7 gene expression and TSWV acquisition of Frankliniella occidentalis.

(A) The experimental procedure for RNAi in Frankliniella occidentalis with TSWV. The blue arrows refer to the entire experimental procedure, and the orange arrows indicate the source of the plant leaves during the experiment. All beans were virus-free. (B) The relative expression levels of UBR7 gene in Frankliniella occidentalis after RNAi at different times. (C) The relative abundance of TSWV in Frankliniella occidentalis after RNAi at different times.

Experiments concerning TSWV acquisition

Collected L1 thrips were divided into three groups (CK, ds-EGFP, and ds-UBR7) of 50 thrips each and were fed an artificial diet for 24 h. Subsequently, thrips were transferred into glass pots and were fed with the leaves of D. stramonium plants carrying TSWV. After 48 h, the leaves were replaced with healthy beans, and the thrips continued feeding until they reached L2. The TSWV abundance of L2 thrips was detected using RT-qPCR. F. occidentalis can only acquire poison in larva, the younger the instar, the greater the ability to acquire poison (Wetering, Goldbach & Peters, 1996). So, detecting TSWV abundance in larval thrips by RT-qPCR can indirectly indicate the ability of thrips larvae to acquire TSWV. The experimental process is shown in Fig. 2A.

Figure 2 Effect of RNAi on acquiring or transmitting TSWV of Frankliniella occidentalis.

(A–D) CK: 30% sucrose + ddH2O, blank control; ds-EGFP: 30% sucrose + dsRNA-EGFP, negative control group; ds-UBR7: 30% sucrose + dsRNA-UBR7, experimental group. (A) The experimental procedure of F. occidentalis to acquire TSWV. (B) The ability of F. occidentalis to acquire TSWV after RNA interference. After RNAi, the abundance of TSWV in F. occidentalis was detected by RT-qPCR, which served as an indicator of the ability of F. occidentalis to acquire TSWV. Again, Actin was used as the reference gene. (C) The experimental procedure of F. occidentalis to transmit TSWV. (D) The relative absorbance of TSWV in leaf discs. After RNAi, the abundance of TSWV in small leaf discs was measured by ELISA, which served as an indicator of the ability of F. occidentalis (V+) to transmit the TSWV. (A, C) The blue arrows refer to the entire experimental procedure, and the orange arrows indicate the source of the plant leaves during the experiment. All beans were virus-free. Values (means ± S.E.) represent data obtained in three independent experiments (n = 3). The asterisk indicates a significant difference according to an independent-sample t-test (NS = Not Significant; ***P < 0.001; ****P < 0.0001).

Experiments for TSWV transmission

The entire experimental procedure for assessing the transmission efficiency of thrips inoculated with TSWV using the leaf disc test was based on previous experimental methods with some improvements (Jacobson & Kennedy, 2013; Okazaki et al., 2011; Okuda et al., 2013). Collected L1s were allowed to feed on TSWV-infected D. stramonium leaf tissue for 72 h. Then, thrips were transferred to cups with green beans. Adult thrips with TSWV 3 days after eclosion were collected and divided into three groups (CK, ds-EGFP, and ds-UBR7) with 50 thrips in each group fed an artificial diet for 24 h. The spread of TSWV is influenced by the sex, age and viral load of the vector insects (Wan et al., 2020). To reduce experimental error, we selected thrips by random selection, regardless of thrips’ sex, to ensure consistent thrips sex ratios and consistent viral loads in each group. Later, the artificial diet was removed from the tube and replaced with a true leaf of a healthy D. stramonium, a small disc with a diameter of 5 mm. Three small disks were placed in each small cage. After 48 h, the small disks were removed and placed in a 24-well plate; 1 mL ddH2O was added to each well, and the plates were placed in an illumination incubator for 72 h. A double-antibody sandwich enzyme-linked immunoassay method was then used to detect the TSWV infection rate of the leaves. The process workflow is shown in Fig. 2B.

In the next step, 100 μL of Phosphate buffered saline (PBS) was used to grind and break the small disks from the above Methods 6: 6. Experiments concerning TSWV acquisition. The samples were centrifuged at 10,000 × g for 5 min, and the supernatant was transferred to a new tube for TSWV enzyme-linked immunosorbent assay (ELISA) (Mmbio, Jiangsu, China) (Jacobson & Kennedy, 2013). Then the absorbance of the sample was measured under the enzyme label instrument (BioTek, Winooski, VT, USA). The absorbance indirectly reflected the ability of thrips to transmit TSWV.

Real-time qPCR

The expression levels of UBR7 were examined using RT-qPCR. First, primer sequences were designed using Primer Premier 5.0 software or a reference research report (Table 1) (Wu et al., 2018; Yuan, 2019). Then, according to the instructions of a 2 × T5 Fast qPCR Mix (SYBR Green I) kit (Tsingke, Beijing, China), RT-qPCR amplification (n = 6) was conducted in a PCR system using a 20 μL reaction volume containing 10 μL 2 × T5 Fast qPCR Mix (SYBR Green I), 0.8 μL of primers (10 μM) and ~50 ng of cDNA. The reaction conditions were as follows: initial denaturation (1 min at 95 °C) followed by 45 amplification cycles of 95 °C for 10 s and 58 °C for 60 s, and the fluorescence signal value was obtained at 60 °C for 1 min. Melt curves were generated to confirm that only one specific PCR product was amplified and detected. The relative gene expression levels were calculated using the 2−ΔΔCT method. Actin was the reference gene (Wu et al., 2018; Yuan, 2019).

Western blotting analysis

Total protein was extracted from thrips with a sodium dodecyl sulfate sample buffer. The concentration of loading proteins was measured by Epoch™ Multi-Volume Spectrophotometer System (BioTek, Winooski, VT, USA) and normalized. Western blotting was performed using anti-β-actin (TransGen, Beijing, China) and anti-UBR7 antibodies (AtaGenix, Hubei, China).

Anti-UBR7 was prepared by predicting the antigenic determinant through BepiPred-2.0, synthesizing the peptide (Peptide 1: CKRPYPDPEDTSDDE; Peptide 2: Cys+NTPGSSSQKSNIETP), and then preparing the antibody. The antibody-reactive bands were revealed using enhanced chemiluminescence (Beyotime, Shanghai, China) and detected using photographic film. Then, in strict accordance with the software operation instructions, the intensity of the bands was quantified using Quantity One v4.6.2.

Surface plasmon resonance technology analysis

SPR is an optical biosensing method that can directly determine the biomolecular interaction kinetic parameters, enabling the monitoring of biomolecular interactions and the quantification of biomolecules in a label-free manner (Homola, Yee & Gauglitz, 1999; Schasfoort, 2017). It is widely used in medical diagnostics, pharmaceutical and food research fields (Wang et al., 2018). According to bioinformatic analysis, UBR7 has two domains, a zinc finger domain of 51–115 AAs and a PHD-SF superfamily of 142–193 AAs, and 51–193 AAs were selected as the target fragment for expression. First, the UBR7-domain sequences were synthesized by Tsingke Biotechnology after optimizing the nucleotide sequences by Escherichia. coli (E. coli) codon preferences. Then, the target fragment was subcloned into the pET28b+ vector E. coli to express the target protein, and a 6 × His tag was added to the N-terminus for expression and purification. The purified UBR7-domain protein and Nicotiana benthamiana (N. benthamiana) with TSWV were sent to AtaGenix for the SPR test. The protein eluate was analyzed with liquid chromatography tandem mass spectrometry (LC-MS/MS), QE by BGI Genomics. Purified UBR7-domain proteins were stored at −80 °C for downstream experiments such as GST pulldown assays.

GST pull-down assay

GST pull-down is a well-established method to detect protein interactions in vitro and can be used to verify the possible direct interaction of two known proteins. The target fragment TSWV N was subcloned into the pGEX-6P-1 vector using the full-length cloning plasmid of the TSWV N gene as a template. E. coli expressed the target proteins TSWV N and GST; an N-terminal GST tag was added for expression and purification. The purified TSWV N-GST and GST proteins interacted with the UBR7-domain-His protein through BeyoMag™ Anti-GST Magnetic Beads (Beyotime, Shanghai, China). The input and pull-down results were analyzed using Western blotting after incubation with anti-UBR7 and anti-GST (Beyotime, Shanghai, China).

Co-immunoprecipitation analysis

Co-immunoprecipitation (Co-IP) was conducted according to the manufacturer’s instructions for Protein A+G Magnetic Beads (Beyotime, Shanghai, China). Briefly,the tissue lysates of F. occidentalis with TSWV were incubated overnight with anti-lgG and anti-TSWV N in rotation at 4 °C. The lysates with primary antibody were then mixed with Protein A+G magnetic beads and incubated with rotation at 4 °C for 3 h. Bead containing immunoprecipitated proteins werewashed 5 times with ice-cold PBS and boiled with 1 × SDS loading buffer. The supernatant was then subjected to immunoblotting analysis with anti-TSWV N and anti-UBR7.

Statistical analysis

All statistical results were expressed as the mean ± standard error (means ± S.E.) of at least three independent experiments. In the following steps, all data were processed with SPSS version 19.0 (SPSS Statistics for Windows, Chicago, IL, USA) after confirming data normality and homogeneity of variances. Differences in relative gene expression were determined by one-way analysis of variance (ANOVA) with Duncan’s multiple range test. Treatments not sharing a common letter in graphs significantly differed at P < 0.05. Alternatively, independent-sample t-tests were used to compare the differences between the two groups (*, P < 0.05; **, P < 0.01; ***, P < 0.001; ****, P < 0.0001).

Results

Differential gene expression associated with virus infection

Transcriptome sequences of F. occidentalis response to TSWV were downloaded from NCBI. After analysis, the top twenty most responsive transcripts in each stage and their Blastx annotations are shown in Table S2. Within the top twenty genes responding to virus infection, all were down-regulated in larvae, while the majority were up-regulated in the pupae and adult stages. The top three differentially expressed genes were TCONS_00003267, FOCC017110-RA, and FOCC003013-RA, and the fold changes were 17.86, 15.38, and 13.31, respectively. Both TCONS_00003267 and FOCC017110-RA had no significant Blastx annotations (NA). Excluding genes with NA, FOCC003013-RA was the most up-regulated gene. The GO processes involved in FOCC003013-RA were molecular function: metal ion binding, oxidoreductase activity, monophenol monooxygenase activity; biological process: metabolic and oxidation-reduction processes.

The amino acid sequence of FOCC003013-RA with 314 amino acids, was compared by NCBI BLAST analysis, and it was found to be highly similar to PREDICTED: Frankliniella occidentalis putative E3 ubiquitin-protein ligase UBR7 (NCBI Reference Sequence: XM_026422690.1). The E-value was 0, and the percent identity was 99.68%. Therefore, primers were designed according to the sequence of XM_026422690.1 to amplify the full-length sequence of the UBR7 open reading frame (ORF) for subsequent research.

cDNA cloning and characterization

After sequencing, the full-length coding sequence of F. occidentalis UBR7 was obtained (Fig. 3). The UBR7 gene contained an unbroken open reading frame of 1,246 nucleotides, encoding 414 amino acids. The amino acid sequence of UBR7 was compared with that of FOCC003013-RA, and there were only two site differences in the sequences of the first 300 amino acids. Based on the deduced AAs, the theoretical molecular weight of UBR7 was predicted to be 46.55 kDa with a theoretical pI of 4.89 (Table S3). Protein BLAST and identity analyses suggested that the UBR7 gene belongs to the E3 ubiquitin–protein ligase family that recognizes N-degrons and structurally related molecules for ubiquitin-dependent proteolysis or related processes through the UBR box motif. In addition, UBR7 harbors a PHD finger domain (142–193 AAs) and a conserved zinc finger domain (51–115 AAs) (Fig. 3), which can specifically recognize histone modifications and some DNA sequences that participate in plant life processes including plant autoimmunity (Mouriz et al., 2015). Therefore, we suspect that F. occidentalis UBR7 is similar to UBR7 in plants. Compared to F. occidentalis UBR7-domains (51–193 AAs) within TSWV-susceptible plants, the UBR7 AAs in this region were highly conserved (Protein sequence identity NaN% = 0.56). The larger the letter in Fig. 4, the more conserved the amino acid site. Within the neighbor-joining phylogenetic tree, F. occidentalis UBR7 belonged to “the E3 ubiquitin–protein ligase UBR7 F. occidentalis” clade, It was clustered with T. palmi, C. secundus, and P. humanus corporis in the same branch (Fig. 5).

Figure 3 Nucleotide and deduced amino acid sequence of the UBR7 cDNA in Frankliniella occidentalis.

The sites of the conserved motifs are marked with a box. The EF-hand calcium-binding domain is shaded. The asterisk indicates the termination codon.

Figure 4 Multiple sequence alignment of Frankliniella occidentalis UBR7-domino amino acid sequences with Solanaceae plants.

The same background color indicates conserved amino acid residues. Gaps (–) were introduced into the sequences to optimize alignment. Small black squares: gene absence/presence variations. Small brown squares: conservation weights.

Figure 5 Phylogenetic analysis of UBR7 amino-acid sequences in Frankliniella occidentalis.

The neighbor-joining phylogenetic tree illustrates the phylogenetic relationship of UBR7 in F. occidentalis with other arthropod E3 ubiquitin–protein ligases. The neighbor-joining method generated the phylogenetic tree with 1,000 bootstrap replicates using MEGA 5.0 according to the amino acid sequences. The numbers above the branches show support from amino acid sequences, and only values above 40% are shown. The tree was drawn to scale, wherein branch lengths were in the same units as those of the evolutionary distances used to infer the phylogenetic tree.

Expression of UBR7 in F. occidentalis

To investigate whether the expression of the UBR7 gene in F. occidentalis (V−) was related to the developmental period, RT-qPCR and Western blotting were used to detect the expression levels of the UBR7 in L1s, L2s, pupae, female adults, and male adults. The protein expression results were consistent with the mRNA levels. The expression levels in the adult stages were significantly higher than those in other stages, with no gender difference (Figs. 6A and 6B). Furthermore, there was no significant difference between the L1s, L2s, and pupal stages. Meanwhile, there was no significant difference in the expression of the UBR7 between female and male adults, but there were substantial differences compared with other instars. This result indicated that UBR7 may be more involved in its biological functions during the adult stage of thrips.

Figure 6 Expression of UBR7 gene in Frankliniella occidentalis.

(A, B) The relative gene expression of UBR7 in different instars (L1s, L2s, pupae, female, and male) of F. occidentalis (V−). (C, D) The relative gene expression of UBR7 in adult F. occidentalis with TSWV (V+) or without TSWV (V−). (E, F) The relative gene expression of UBR7 in different tissues of adult F. occidentalis. (A, C, E) Actin was used as a reference gene in the RT-qPCR. (B, D, F) β-actin was used as an internal reference protein in Western blotting. Left panel: Western blotting images; Right panel: quantification of Western blotting images. Values (means ± S.E.) represent data obtained in three independent experiments (n = 3). Treatments not sharing a common letter are significantly different at P < 0.05 as assessed by one-way ANOVA followed by Duncan’s test.

The transmission ability of male western flower thrips was higher than that of F. occidentalis (Ogada & Poehling, 2015; Stafford, Walker & Ullman, 2011). E3 ubiquitin-protein ligases are associated with the infection of hosts by viruses and transmission of viruses by insect vectors (Snippe, Goldbach & Kormelink, 2005). Therefore, we questioned whether the expression of UBR7 varies depending on the sex of F. occidentalis. To explore whether TSWV influences the UBR7 gene of F. occidentalis in different sexes, we used RT-qPCR and Western blotting to detect the expression level of UBR7 in female and male thrips with and without TSWV (Figs. 6C and 6D). The results of UBR7 at the mRNA transcript level were generally consistent with those at the protein level. When male and female thrips did not carry TSWV (♀V−, ♂V−), there was no significant difference in UBR7 expression. There were also no significant differences between sexes when exposed to TSWV when both male and female thrips carried TSWV (♀V+, ♂V+).

To investigate whether UBR7 was differentially expressed in different insect tissues, we divided F. occidentalis into three parts: head, thorax, and abdomen, and examined the relative expression of UBR7 by RT-qPCR and Western blotting (Figs. 6E and 6F). The protein detection results were consistent with the changes in mRNA levels. UBR7 was highly expressed in thoracic tissue and was significantly differentially expressed compared to its expression in the head and abdomen.

Effects of RNAi on F. occidentalis

Effects of RNAi on UBR7 gene expression in F. occidentalis without TSWV

Compared with that in the CK and ds-EGFP groups, the UBR7 gene expression in the ds-UBR7 experimental group was significantly reduced (Fig. 7A). Furthermore, the result indicated that this dsRNA fragment could effectively down-regulate the expression of the UBR7 gene in thrips (V−).

Figure 7 Effects of UBR7 RNA interference on Frankliniella occidentalis.

The thrips were fed the artificial diet containing dsRNA. Twenty-four hours later, the interference efficiency of the UBR7 gene in F. occidentalis was examined by RT-qPCR, and the survival rate of F. occidentalis was determined. (A) The relative gene expression of UBR7 in F. occidentalis (V−) after RNAi. Actin was used as the reference gene. (B) F. occidentalis (V−) survival rate after RNAi. (survival rate = the number of live thrips after interference/the number of live thrips before interference). Values (means ± S.E.) represent data obtained in three independent experiments (n = 3). The asterisks indicate significant differences according to independent-sample t-tests (*P < 0.05).

Effects of RNAi on the survival rate of F. occidentalis without TSWV

To understand whether UBR7 affected the vital activity of thrips, we recorded the survival rate of thrips 24 h after RNAi feeding. Compared with the CK group and the ds-EGFP group, the survival rate of F. occidentalis (V−) in the ds-UBR7 group decreased significantly (Fig. 7B). Therefore, it was hypothesized that the UBR7 gene might be involved in F. occidentalis (V−) vital activity.

Effect of RNAi on UBR7 gene expression and viral abundance in F. occidentalis with TSWV

RNAi was performed on adult thrips carrying TSWV (Fig. 1A). As the time of RNAi was extended, the expression of the UBR7 gene gradually decreased and then slowly recovered, with an overall U-shape pattern (Fig. 1B). Similarly, the abundance of TSWV was consistent with the expression of UBR7, which also decreased and then recovered (Fig. 1C). This implied a link between the expression of UBR7 and the abundance of TSWV in thrips with TSWV.

Effect of RNAi on acquiring TSWV of F. occidentalis

To explore the effect of UBR7 on the ability of thrips (V−) to acquire TSWV, RT-qPCR was used to detect TSWV in thrips larvae after feeding on leaves containing the virus. Compared with those in the CK group and the ds-EGFP group, there was no significant difference in the ability of F. occidentalis in the ds-UBR7 group after RNAi. Fig. 2B shows that UBR7 did not affect the ability of F. occidentalis to acquire TSWV. In this study, UBR7 was only highly expressed in adult stages and at low levels in the larval stage (Figs. 6A and 6B), indicating that it was less important in the larval stage.

Effect of RNAi on transmitting TSWV of F. occidentalis

UBR7 was highly expressed in the adult stage of F. occidentalis (Figs. 6A and 6B), and E3 ubiquitinase is related to the spread of the virus (Snippe, Goldbach & Kormelink, 2005). Therefore, we used ELISA to detect the protein concentration of TSWV in leaves to explore the effect of UBR7 on the ability of F. occidentalis (V+) to transmit TSWV. After RNAi with F. occidentalis (V+), the absorbance value in the ds-UBR7 group (experimental group) was significantly lower than that in the CK group (blank control) (P < 0.0001) and ds-EGFP group (negative control group) (P < 0.001) (Fig. 2D). This result demonstrated that down-regulation of the UBR7 gene impaired the ability of thrips to transmit TSWV.

UBR7 protein interacts directly with the TSWV N protein

The proteins interacting with UBR7-domain were screened using SPR and LC-MS/MS and then compared with the NCBI datasets. The results showed the top five proteins with the highest scores (Table 2). Most matched proteins were from the viral host N. benthamiana and viruses in the genus Orthotospovirus such as TSWV and Chrysanthemum stem necrosis orthotospovirus. Among these, the highest score was the N protein of TSWV (GenBank accession number: gi|284810746); this explained why UBR7 bound to the TSWV N protein with high efficiency. Detailed peptide information can be found in Figs. S6–S9.

Table 2 The top five most strongly interacting proteins with UBR7-domino.

Accessiona	Description	Mass	Score	Matchesb	Sequencesc	emPAI	Coverage	
gi|284810746	nucleocapsid protein [Tomato spotted wilt orthotospovirus]	29,094	650	34 (23)	16 (12)	4.08	51%	
gi|729042213	glyceraldehyde 3-phosphate dehydrogenase-A [Nicotiana benthamiana]	42,945	115	10 (5)	7 (5)	0.45	19%	
gi|926663240	heat shock protein 90-1 [Nicotiana benthamiana]	80,443	94	6 (3)	5 (2)	0.08	6%	
gi|1219878403	Gc-Gn glycoprotein precursor [Chrysanthemum stem necrosis orthotospovirus]	130,189	89	6 (4)	5 (4)	0.10	4%	
gi|660450867	domains rearranged methyltransferase 1 [Nicotiana benthamiana]	69,106	78	4 (4)	1 (1)	0.05	0%	
Notes:

a The protein number in NCBI.

b The total number of peptides matched, in brackets is the number of matches higher than the significance threshold.

c The total number of sequences matched, in brackets is higher than the significance threshold sequence number.

To further verify the results of SPR and LC-MS/MS, GST pull-down and Co-IP assays were used to analyze the interaction between UBR7-domain and TSWV N. In the GST pull-down assay, TSWV N, GST and UBR7 could be detected in the input (Fig. 8A), indicating that the GST pull-down system was functional. Fig. 8B shows the result after the beads pull down. UBR7-domain could be detected after co-incubation with TSWV N-GST but not after co-incubation with GST (Fig. 8B). In the Co-IP assay (Fig. 8C), TSWV N and UBR7 could be detected in the input, meaning the Co-IP system usually works. After immunoprecipitation, no bands existed in the lgG-negative control group, while TSWV N and UBR7 bands could be detected in the TSWV N group. In brief, GST pull-down and Co-IP confirmed the direct interaction between TSWV N and UBR7 in vitro and in vivo, respectively, which was consistent with the results of SPR.

Figure 8 The UBR7 protein in Frankliniella occidentalis interacts directly with the nucleocapsid protein of TSWV.

(A, B) The GST pull-down assay examined the interaction between UBR7 and TSWV N in vitro. Anti-GST and anti-UBR7 antibodies were used to test input and pull-down samples. (A) Input proteins before the GST-bead pull-down, TSWV N and UBR7 could be detected as the signal, indicating that the GST pull-down system was functional. (B) Pull-down proteins after the GST-bead pull-down. UBR7 can be seen in UBR7-domain-His co-incubated with the TSWV N-GST group and UBR7-domain-His group, but not in GST co-incubated with UBR7-domain-His group, indicating that the UBR7-domain specifically interacted directly with the TSWV N protein in vitro. (C) The Co-IP assay examined the interaction between UBR7 and TSWV N in vivo. Anti-TSWV N and anti-UBR7 antibodies were used to test Input and IP. Input: complete protein extract. IP: immunoprecipitated proteins. TSWV N and UBR7 can be detected at Input, indicating that the system works properly. After immunoprecipitation, there were no bands in the lgG negative control group, while TSWV N and UBR7 bands were detected in the TSWV N group. Co-IP confirmed the interaction between TSWV N and UBR7 in vitro and in vivo.

Discussion

Ubiquitin is an essential protein in PTM widely involved in regulating innate immune signaling pathways. Ubiquitin is highly conserved in all eukaryotes (Alejandro et al., 2019; Heaton, Borg & Dixit, 2016; Schinz & Littlefield, 1985). Ubiquitination regulates protein homeostasis, cell cycle progression, gene transcription, receptor transport, and immune response (Haglund & Dikic, 2014). Ubiquitination regulation is not limited to eukaryotic cells but is also found in viruses (Gao et al., 2021; Liu, Cai & Gao, 2018). Many viral proteins can mimic or usurp key regulators that affect the binding of ubiquitin-like protein modifiers and ubiquitin-proteins in the host and interfere with the corresponding enzymatic cascades to effectively promote viral replication (Gao et al., 2021; Stukalov et al., 2021). For the virus–host interaction, previous studies have shown that the virus has evolved strategies to exploit specific PTM processes during the immune evasion (Wimmer, Schreiner & Dobner, 2012).

Through NCBI sequence alignment analysis, the highly expressed gene FOCC003013-RA in F. occidentalis response to TSWV belongs to PREDICTED: Frankliniella occidentalis putative E3 ubiquitin-protein ligase UBR7 (NCBI Reference Sequence: XM_026422690.1), and we also named it UBR7. In the previous study, FOCC003013-RA was described as a hemocyanin subunit type 1 precursor. However, when the amino acid sequences of UBR7 and FOCC003013-RA were compared with the “hemocyanin subunit 1 precursor” of insect origin, we found that the amino acid sequences of UBR7 and FOCC003013-RA showed very little consistency with other sequences and no conserved structural domain of hemocyanin subunit 1 precursor (Fig. S8). UBR7 belongs to the E3 ubiquitin–protein ligase family with typical RING-type family characteristics and encodes a UBR box-containing protein, with a PHD in the C-terminus and a zinc finger in the N-recognin (Figs. 3 and 5) (Dasgupta et al., 2022). Also, plant E3 ubiquitin–protein ligase is essential in regulating hormone responses, morphogenesis, disease resistance, and abiotic stress response. Overexpression of the plant E3 ubiquitin–protein ligase BnTR1 in rapeseed and rice enhanced the resistance to heat stress (Liu et al., 2013). CaRING1 silencing increased the infection rate of plants to bacterial spots in red pepper (Dong, Choi & Hwang, 2011). The UBR7 protein in the E3 ubiquitin–protein ligase family in tobacco directly interacted with the N TIR-NLR, and UBR7 downregulation enhanced TMV resistance (Zhang et al., 2019). Thus, we hypothesized that if UBR7 of F. occidentalis also has a similar immune function to plant UBR7, their amino acid sequences have certain similarities. By comparing AAs, we confirmed the high similarity between F. occidentalis and Solanaceae plant UBR7, which made us more confident on our speculation (Fig. 4). Therefore, we hypothesized that the UBR7 protein in F. occidentalis may be closely related to TSWV. SPR (Table 2, Figs. S6–S9), GST pull-down and Co-IP experiments (Fig. 8) all demonstrated that western flower thrips UBR7 interacts with the TSWV N protein, validating our hypothesis.

Most of the current research on thrips interactions with TSWV have focused solely on TSWV (Gupta, Kwon & Kim, 2018; Montero-Astúa et al., 2014), while studies on thrips have stayed at the ecological (Gupta, Kwon & Kim, 2018; Sarwar, 2020; Stumpf & Kennedy, 2005), transcriptomic (Schneweis, Whitfield & Rotenberg, 2017), proteomics levels (Badillo-Vargas et al., 2019; Zheng et al., 2020). There are few studies on the biology and gene function of TSWV affecting thrips (Badillovargas et al., 2012). In F. occidentalis, the high expression of the UBR7 gene primarily occurred in the adult stage, and UBR7 (FOCC003013-RA) was also the most up-regulated gene in transcriptome data; these all indicated the temporal specificity of UBR7. Given the differences between the transcriptome and proteome at the tissue level in the larval and adult stages, and the fact that F. occidentalis can only transmit TSWV in adult stages after the thrips acquire the virus in the larval stages (Wetering, Goldbach & Peters, 1996), we again hypothesized that UBR7 is strongly associated with TSWV transmission in F. occidentalis. In addition, TSWV induced high expression of UBR7 in F. occidentalis (Figs. 6C and 6D), mainly in the thorax of the thrips (Figs. 6E and 6F). Arboviruses of plants circulate and multiply within the insect vector; the vector serves as an alternative host for the plant pathogen (Geetanchaly & John, 2021). Usually, the vector acquires the plant pathogen by feeding on infected plants. It has been hypothesized that once the virus enters the insect, it will cross intestinal barriers, internal organs, and visceral muscles and can be found throughout the hemolymph (Perilla-Henao & Casteel, 2016). The salivary gland is an essential organ for the persistent transmission of viruses in insect vectors; it is the last line of defence that plant viruses need to overcome to circulate in the insects (Stafford-Banks et al., 2014). For vector insects to transmit pathogens to new plant hosts, the virus must spread from the hemolymph to the salivary glands (Whitfield, Falkb & Rotenberga, 2015). At the same time, salivary glands are found mainly in the thorax of insects (Hiroaki et al., 2020; Ohnishi et al., 1996). TSWV circulates and proliferates in F. occidentalis and is spread by adult thrips (Gupta, Kwon & Kim, 2018). Therefore, UBR7 was highly expressed in the thoracic tissue of F. occidentalis. We hypothesize that this was because TSWV was enriched in the salivary glands of adult thrips, inducing high expression of UBR7.

Salivary glands, one of the essential organs of vector-borne viruses, are closely related to virus transmission. Thus, we speculated that UBR7 participates in transmitting TSWV by F. occidentalis. From this, we decided to measure changes in the transmission of F. occidentalis after RNA interference. The transmission of TSWV can be influenced by host plants, virus load, sex, age, susceptibility to pesticides, and behavior of its insect vector, thrips (Kumm & Moritz, 2010; Maris et al., 2004; Pappu, Jones & Jain, 2009; Stafford, Walker & Ullman, 2011; Whitfield & Oliver, 2016; Zhang et al., 2015; Zhao et al., 2016). Carrying the virus could improve the reproduction rate of F. occidentalis (Maris et al., 2004) and affect the sex ratio of thrips, increasing the number of male thrips, sex with a greater dispersal and virus transmission capability (Wan et al., 2020). At the same time, TSWV could also alter the feeding behavior of F. occidentalis, in which virus carriers fed significantly more than the non-carriers (Stafford, Walker & Ullman, 2011).

As a result, the transmission efficiency of F. occidentalis varies with gender (Liao et al., 2015). In the field, we could not determine the number of thrips carrying TSWV in thrips populations. There is also no way to determine the virus load in each thrips. Therefore, to reduce experimental error due to differences in sex and individual viral load, we selected thrips at random, regardless of sex, to ensure consistent sex ratios and consistent viral loads in each group of thrips. The results of RNAi experiments confirmed our hypothesis, in which the down-regulation of UBR7 attenuated TSWV transmission. Simultaneously, UBR7 expression decreased and then increased with increasing RNAi time. This was caused by the gradual degradation of dsRNA, which can no longer interfere with the UBR7 gene. And we did not add additional artificial feed containing dsRNA throughout the RNAi experiment. As UBR7 expression recovered, the abundance of TSWV also started to rise, presumably because UBR7 influenced the proliferative effect of TSWV and thus reduced the ability of western flower thrips to transmit the virus. The abundance of the virus in vector insects is also a critical factor in the success of TSWV transmission by thrips Nagata et al. (2002). Rotenberg and colleagues found that virus titer was positively associated with the frequency of transmission events (Rotenberg et al., 2009). With low URB7 expression, the efficiency of TSWV transmission by thrips decreased.

Meanwhile, RNAi had little effect on the ability of thrips to acquire TWSV. The typical expression of UBR7 in the larval stage suggests that it has a minor physiological role in the larval stage. Thrips acquire the virus at L1 and L2 (Wetering, Goldbach & Peters, 1996), with a lower expression of UBR7 in the instar stages. As the expression level of UBR7 is low in the thrips larval stage, RNAi does not significantly affect UBR7 expression, so it will not have a significant biological impact. Therefore, we speculate that UBR7 could interfere with virus replication and thus influence the transmission efficiency of F. occidentalis, while the efficiency of TSWV acquisition was unaffected. The movement of the virus from the salivary glands of thrips to the new host plant may not be affected by protein silencing, which still requires further investigation.

Meanwhile, many proteins interacting with UBR7-domain were screened by SPR (Table 2), among which TSWV N was the most closely involved. At the same time, we verified the direct interaction between UBR7 protein and TSWV N by GST pull-down (Figs. 8A and 8B) and Co-IP (Fig. 8C) in vitro and in vivo respectively. It was proved that the UBR7 domain site (51-193AA) in F. occidentalis can directly interact with the N protein of TSWV. These results are consistent with our initial hypothesis that TSWV N directly interacts with UBR7.

The N protein is a phosphorylated nucleocapsid protein of the virus, and the epitope of N protein can induce the body to produce an effective immune response (Shi et al., 2017; Zhou et al., 2019). N protein also protects internal nucleic acids from damage by nucleases in the external environment (Afrasiabi et al., 2020). UBR7 is highly expressed in adult thrips carrying TSWV (Figs. 6A–6D). We hypothesize that RNAi reduced the expression of UBR7, thus preventing the interaction between UBR7 and TSWV-N, breaking the homeostasis in thrips, and resulting in increased mortality, reduced activity and decreased transmission ability of thrips. However, the exact mechanism still needs to be further explored and verified.

Conclusions

In conclusion, our study found that the UBR7 gene was involved in TSWV transmission by thrips. The results demonstrated the direct interaction between the F. occidentalis UBR7 protein and the TSWV N protein (51–199 AAs domain). UBR7 was expressed and functional in the adult stage, mainly in thrips’ thoracic tissue. Down-regulation of UBR7 expression reduced the transmission efficiency of thrips. These results suggested that UBR7, an E3 ubiquitin–protein ligase family member, is closely related to TSWV transmission by F. occidentalis. The pathway through which the UBR7 protein affects TSWV proliferation needs further characterization. The molecular structure where the UBR7 protein interacts with TSWV N in F. occidentalis remains to be studied. Moreover, we will continue to explore whether UBR7-targeted pesticides will help TSWV transmission and F. occidentalis control. Our study provides insight into the mechanism of TSWV transmission by F. occidentalis.

Supplemental Information

Supplemental Information 1 Supplementary Figures, Tables 1–8, 10 and 11 and Information.

Click here for additional data file.

Supplemental Information 2 After surface plasmon resonance, the peptide sequence was identified by LC-MS/MS, QE by BGI Genomics.

Click here for additional data file.

Supplemental Information 3 Raw film from western blot experiment (Fig. 5) and GST pull down (Fig. 8).

Click here for additional data file.

Supplemental Information 4 The raw data for Figs. 1, 5–7.

Each values are representative of data obtained from at least three independent experiments (n ≥ 3).

Click here for additional data file.

Supplemental Information 5 Down-regulation of the UBR7 gene reduced the efficiency of thrips in transmitting TSWV.

Click here for additional data file.

We thank the Institute of Vegetables and Flowers, Chinese Academy of Agricultural Sciences, for the practical technical guidance. In addition, we thank LetPub for its linguistic assistance while preparing this manuscript.

Additional Information and Declarations

Competing Interests

Author Contributions

Data Availability

The authors declare that they have no competing interests.

Junxia Shi conceived and designed the experiments, performed the experiments, analyzed the data, prepared figures and/or tables, authored or reviewed drafts of the article, and approved the final draft.

Junxian Zhou performed the experiments, prepared figures and/or tables, and approved the final draft.

Fan Jiang conceived and designed the experiments, prepared figures and/or tables, and approved the final draft.

Zhihong Li conceived and designed the experiments, authored or reviewed drafts of the article, and approved the final draft.

Shuifang Zhu conceived and designed the experiments, authored or reviewed drafts of the article, and approved the final draft.

The following information was supplied regarding data availability:

The raw measurements are available in the Supplemental Files.

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
