# Peer review of "The effects of the E3 ubiquitin–protein ligase UBR7 of Frankliniella occidentalis on the ability of insects to acquire and transmit TSWV"

_PeerJ, doi:10.7717/peerj.15385_

## Round 0.1 · original submission · Major Revisions

Please provide a comprehensively revised version addressing the editorial comments and a detailed rebuttal letter.
Key aspects that need to be addressed in the modified manuscript and explained in the letter are:

-It appears that the RNA-seq dataset was taken from GenBank, therefore, how it can be used for this study? How useful is the differential expression obtained from those dataset related to the experimental word reported here?
An experimental validation using RT-qPCR of up-regulated and down-regulated genes would be a way to validate the use of third-party data, which was not done.

The only validation done was that the gene silencing occurred.

The current manuscript also lacks information and discussion about how the replication mechanisms are modified and how the virus is transmitted. Also, I share the concerns of the reviewers about generalizations made from other plant-virus interactions that are not supported by your work with TSWV.

Please be encouraged by the extensive revisions, comments and suggestions, we look forward for your revised manuscript.

Reviewer 1 ·

Basic reporting

The effects of the E3 ubiquitin–protein ligase UBR7 of Frankliniella occidentalis on the ability of insects to acquire and transmit TSWV (#80617)
The authors tested the hypothesis that E3 ubiquitin ligase, UBR7 was involved in the molecular interactions between thrips and TSWV. Expression of UBR7 was tested in different tissues and developmental stages to characterize the constitutive expression of the transcript. Additionally, the expression of UBR7 was tested upon exposure to TSWV both at transcript and protein level. Expression of UBR7 was silenced to test the hypothesis of involvement of UBR7 in TSWV infection by RNAi through feeding. The effect of knockdown was tested on both acquisition and transmission of TSWV by larvae and adults respectively. Authors found interesting results wherein the knockdown of UBR7 had an effect on transmission but not acquisition. An SPR revealed that the interacting proteins of UBR7 included TSWV N protein which was tested by GST pull down assay.
Overall, this is very exciting and interesting study given the growing importance of TSWV and thrips interaction in the field of vector biology. Authors have performed the experiments thoroughly with sufficient experimental replicates and these results add valuable knowledge to Thrips-tospo interactions.
However, the manuscript needs significant improvement in writing as there are many details missing particularly in the methods section. Some of the methods for the results included are missing and needs to be included in the sections. Please include a statement highlighting the rationale for performing each experiment to better orient the readers. Sequence of the methods could also be better organized to reflect the flow of the experiments. Another major concern for this study is the random selection of the sexes for the experiments particularly the transmission experiment. Males and females have different transmission efficiency due to their feeding behavior. These would have an effect on the ELISA results from the Datura leaves that were fed upon by the infected thrips from the dsRNA treatments. Authors needs to clarify this detail.
Another major area of improvement that the manuscript needs is in the discussion section. For the interesting results the authors found, the discussion could be much more impactful. Given the availability of the transcriptome and proteome at the tissue level for the larval and adults stages highlighting the molecular mechanisms for acquisition and transmission, the authors could study the behavior of UBR7 in these studies to discuss the results identified in the current study. For example, there is a E3 ubiquitin ligase identified in the saliva of female and male thrips with differential spectral counts with and without TSWV. Similarly, authors can compare the expression of the UBR7 in the gut transcriptome of the larval instars exposed and not exposed to TSWV.
Figure legends need to be more detailed to help the readers interpret the results presented.
Authors need to address these comments before the manuscript can be accepted for publication.

Experimental design

No comment

Validity of the findings

No comment

Annotated reviews are not available for download in order to protect the identity of reviewers who chose to remain anonymous.

Reviewer 2 ·

Basic reporting

On line 51- Franklinella fusca is the scientific name for tobacco thrips.
Also- in line 51- the line, “The western flower thrips (F. occidentalis (Pergande)) is the dominant species…”. Replace ‘dominant’ with ‘most efficient’, there are many that would argue dominance depending on the ecosystem, as in the south of the US, in peanuts, F. fusca is the most prevalent vector for TSWV, not F. occidentalis due to the season that young tissues are present in the fields and the amount of F. fusca present in the ecosystem.
On line 45- TSWV can infect more than 1060 plants in 85 families- Scholthof is not the best reference, it would be better to use Parella et al., 2003 where they actually defined the host range more conclusively than the Scholthof review.

Badillo-Vargas et al., 2019 should be cited as an example of thrips proteins found to be interacting with TSWV, specifically Gn and N. This should be in the same part of the introduction as the Wan et al., reference at line 93.

Line 240- Theys should be they.

As you reassembled an existing dataset already published, it would be appropriate to discuss if this paper’s findings agree with the original published results or if this represents a difference in assembly or in reporting. That information should appear in the discussion.

Line 493- “Plant viruses circulate and multiply within the insect vector…”. This is an incorrect statement- this only applies to a subset of viruses within the Bunyavirales and Rhabdoviridae- the majority of plant viruses DO NOT replicate in their insect vector. This statement needs revision.

Line 497- In the case of TSWV- the virus maintains a low titer in the hemolymph and it is suspected that the mode of movement to the salivary glands is likely through an intestinal route through the Malpighian tubules (Which is the hypothesized reason why the virus must be acquired by larval stages to be transmitted by adults, that the reorganization of the intestinal tracts of the foregut and hindgut are required through pupation to move the virus to the correct location-salivary glands- to be transmitted). The statement in this line is a general statement which is not accurate as a mode for this virus. Suggest revision. The movement through the hemolymph is the most accepted for the begomoviruses which are not propagative in their vectors.

Line 396- The methods states that the RNAi was ingested, however, in this line, you state injection. As injection is the RNAi method introduction of choice in other labs, please clarify this line to match if it is injection or ingestion. If injection- revise methods to reflect this.

Line 501- Whitfield is misspelled.
Line 504- TSWV- “is highly expressed in thoracic tissues?” Are the titer levels high or are the TSWV proteins highly expressed? The virus itself is not expressed, it is either highly replicating or the proteins are high, please clarify.

Line 513- Cite agreement with the Rotenberg et al, 2009 paper here, that when the viral levels are low, no transmission.


Figure 6- and the corresponding methods- need to match- this appears based on methods to be after 24 hours. This should also be stated in figure legend.

Experimental design

What was the rationale for re-assembly of the existing datasets from another publication? Is this due to the accessibility of the final dataset from that paper? Explain why this was done.

Section 5.4- Line 230- specify the age of the thrips in question- how long after adulthood?

Why is feeding used for the RNAi- other labs are using injection, it is not clear why the feeding was used, especially since they specify that after 24 hours the silencing loses effectiveness.

Validity of the findings

Line 527- The excess of N protein causes immune dysfunction and reduced function and survival in thrips. There is no demonstration of any impact on the immune system, this statement only a hypothesis with no evidence and there could be many reasons why this is occurring.

Over and over, the point of this paper appears to be that due to the silencing of UBR7, there is a reduction of transmission of TSWV. Although the data support that, it does not support the mechanism. In Rotenberg et al., 2009 (which is not, but should be, cited), it is demonstrated that the titer levels being too low, caused a reduction in transmission. This point is demonstrated in this paper, that the titer levels are reduced, that TSWV is not replicating to the same levels. It would be better to make the point that UBR7 interferes with the replication of the virus, not specifically the transmission. This is the indirect result of a poorly replicating virus. This is further supported by the interaction with TSWV N which is thought to be part of the replication complex for TSWV. The specific transmission proteins of TSWV are Gn and Gc and this paper does not demonstrate any type of interaction between UBR7 and these proteins. This would suggest that the actual ability to transmit may not be hindered, but the capacity of the virus to replicate to high enough levels for transmission, therefore, making the premise of the paper somewhat misleading. In this system, it has been demonstrated that the first stage is acquisition, then replication and then finally transmission. Either demonstrate that UBR7 interacts with Gn and Gc or re-evaluate your main point and make it clear that the actual transmission mechanisms may not be impacted.

Going along with this concept, the mechanisms of the viral lifecycle should be more clearly presented in the introduction so that readers are aware of the role of TSWV N in replication.

Additional comments

Despite the earlier comments, this reviewer did enjoy the paper. The identification of more interacting partners to any proteins of TSWV is of critical importance to understand how this virus functions in insects. It was frustrating, given the importance, to see the lack of discussion related to the impacts of reduced titer on transmission and no separation between transmission and replication. Although the two processes are linked, if the virus does not reach high enough levels, it is not transmitted. However, it should be stated that the actual transmission, movement from the salivary glands into a new host is likely not impacted by this protein being silenced.

Reviewer 3 ·

Basic reporting

Frankliniella occidentalis is the main vector of TSWV transmission, and there is a close and complex relationship between them. The study of the interaction proteins between them is of guiding significance for the prevention and control of TSWV. The manuscript studied the effects of UBR7 in WFT on the ability to acquire and transmit TSWV. The study makes sense, but it also has some problems.

Experimental design

1. The virus transmission rate of leaf disc method is low, so it is recommended to use live plants for virus transmission verification.
2. There are differences in the amount of virus acquired by individuals of thrips. If the amount of virus acquired by groups is used, it is recommended to conduct three independent experiments to avoid the difference in the amount of virus in the body, which will interfere with the UBR7 gene and lead to no effect in the amount of virus acquired.
3. The interaction between UBR7 and TSWV N is suggested to be verified in vivo, such as yeast two-hybrid, bimolecular fluorescence complementation, and Co-Immunoprecipitation.
In the virus acquisition experiment, the acquisition time of nymphs is 24h. What is the basis for the selection of acquisition time? What is the acquisition rate?

Validity of the findings

1. The biggest problem of this study is that the overall logical relationship is not clear. There are many proteins related to TSWV, so why UBR7 was chosen? What is the reason for choosing UBR7? Moreover, UBR7 interacts with N, and UBR7 is an important protein in the transmission of TSWV. What is the relationship between these two phenomena?
2. The correlation between transmission rate and absorbance should be directly replaced by transmission rate.

Additional comments

My suggestion is that the authors should rearrange the whole paper, clarifies the logical relationship between the experiments, and resubmits the paper.

---

## Round 0.2 · Minor Revisions

Please address in detail the comments by the reviewer and submit a revised version.

Reviewer 2 ·

Basic reporting

The Authors responded to original comments and the paper is much improved. There are minor gramerical issues that should be addressed still, but only one scientific point that remains to be made.

The authors chose not to re-do the analysis borrowed from Genbank and instead use the results as published in order to support the lack of validation by qRT-PCR. As the original dataset was already validated, this seems logical, however, there remains a disconnect as to why FOCC003013-RA was not annotated as they stated in this paper in the original study. Please suggest an explanation why the annotation for this same sequence in the original paper is: hemocyanin subunit type 1 precursor.

How similar is UBR7 to this annotated from the original table?

Experimental design

The experimental design was good, and I agree with the results.

Validity of the findings

This paper was improved by changing the direction of the paper away from strictly transmission to acquisition and replication with an impact on transmission in that manner. It agrees with the findings and is very well done.

Additional comments

This reviewer was mostly satisfied with the changes made, they improved the paper. I would finalize the comparison between the earlier paper as to how this is a new finding. Perhaps a discussion line about how annotations change based on new additions to Genbank informing of better matches than the original data and the need to revisit some data sets again as new information becomes available.

---

## Round 0.3 · Minor Revisions

Thanks for your responses. I have taken the liberty of suggesting some style modifications and correcting some typos. Please check the included PDF and submit a revised version for grammar and style at your convenience.

---

## Round 0.4 · accepted · Accept

Thanks for addressing the minor revisions requested. Now your manuscript is accepted in PeerJ.